# Analysis of Ionic Domain Evolution on a Nafion-Sulfonated Silica Composite Membrane Using a Numerical Approximation Model Based on Electrostatic Force Microscopy

**DOI:** 10.3390/polym14183718

**Published:** 2022-09-06

**Authors:** Osung Kwon, JaeHyoung Park

**Affiliations:** 1Faculty of Science, Tabula Rasa College, Keimyung University in Seongseo, Daegu 42601, Korea; 2Corporate Research Center, HygenPower Co., Ltd., Daegu 42988, Korea

**Keywords:** proton exchange membrane, electrostatic force microscopy, numerical approximation model, proton conductivity, proton transport mechanism, ionic channel distribution

## Abstract

It is important to characterize the proton transport mechanisms of proton exchange membranes (PEMs). Electrostatic force microscopy (EFM) is used to characterize the ionic structures of membranes. In this study, we attempted to quantitatively analyze the proton conductivity enhancement of Nafion-sulfonated silica (SSA) composite membranes with variations in the ionic channel distribution. This study involved several steps. The morphology and surface charge distribution of both membranes were measured using EFM. The measured data were analyzed using a numerical approximation model (NAM) that was capable of providing the magnitude and classification of the surface charges. There were several findings of ionic channel distribution variations in Nafion-SSA. First, the mean local ionic channel density of Nafion-SSA was twice as large as that of the pristine Nafion. The local ionic channel density was non-uniform and the distribution of the ionic channel density of Nafion-SSA was 23.5 times larger than that of pristine Nafion. Second, local agglomerations due to SSA were presumed by using the NAM, appearing in approximately 10% of the scanned area. These findings are meaningful in characterizing the proton conductivity of PEMs and imply that the NAM is a suitable tool for the quantitative assessment of PEMs.

## 1. Introduction

Green energy sources, such as solar and wind energy, are widely accepted as the best ways to decarbonize and solve global warming. Thus, solar and wind energy devices are being used to replace traditional energy systems. However, crucial issues that need to be overcome remain, such as unstable and intermittent energy supplies during the energy conversion process [1]. These issues can be overcome by combining large-scale energy storage systems, such as redox flow batteries and proton exchange membrane fuel cells (PEMFCs). Proton exchange membranes (PEMs) play important roles in PEMFCs, and function as gas separators, providers of electrode insulation, and proton conductors, etc. Moreover, proton conductors are directly related to the performance of PEMFCs.

Nafion, which was developed by DuPont, is one of the most widely used PEMs, due to its suitable proton conductivity and its chemical, mechanical, and thermal characteristics. Nafion has an inhomogeneous structure consisting of hydrophobic fluorinated polymer polytetrafluoroethylene (PTFE) backbones, attached pendant side chains, and sulfonic acid groups (–SO3H) at the side chain terminals. The backbone and sulfonic acid groups create hydrophobic and hydrophilic domains in the membranes [2,3]. The proton conductivity of Nafion is linked to its morphological structure. Gierke et al. [4] studied the morphological structure of Nafion using small-angle X-ray scattering (SAXS) and wide-angle X-ray diffraction (WAXD).

In the cluster network model, sulfonic acid groups create ionic clusters with inverted micellar structures. Spherically shaped hydrophilic ionic clusters with a radius of 2 nm were scattered in a hydrophobic PTFE matrix. Under hydration, ionic clusters become connected to other clusters and produce narrow channels with diameters of 1 nm, which is referred to as an ionic channel network. Protons pass through the ionic channel network by two mechanisms: a vehicle-type mechanism and a Grotthuss-type mechanism [4]. The two mechanisms occur simultaneously when protons move through the membrane. Under wet conditions, the vehicle-type mechanism is dominant, whereas under dry conditions, the Grotthuss-type mechanism is dominant [5].

When developing PEMs, the enhancement of the proton conductivity of PEMs is an essential goal. Therefore, there has been widespread use of novel membranes to achieve productivity enhancements. There are two major types of membranes with enhanced proton conductivities: non-Nafion-based composite membranes [6,7,8] and Nafion-based composite membranes [9,10,11,12]. Nafion-based composite membranes are typically synthesized using inorganic filler materials and Nafion ionomers. Studies of these processes have focused on improving hydrolysis under insufficient water content by using extremely hydrophilic materials and adding sulfonic acid groups to sulfonized filler materials to enhance proton conductivity. Nafion/sepiolite composite membranes were synthesized and characterized by Beauger et al. [13]. In their study, the composite membranes showed a 50% better performance at 100 °C and 50% RH, compared with the performance of pristine Nafion. They explained that the result was due to the impact of the increasing ion exchange capacity and the increasing water diffusion rate on the membrane. Costamagna et al. [14] synthesized Nafion/zirconium phosphate composite membranes for high-temperature PEMFCS. They proposed that the composite membrane showed stable performance at 130 °C, which is a dehydration condition.

Various filler materials have been proposed, including nanostructured silica, graphene, and titanium oxide. These materials are sulfonated and combined with a Nafion ionomer to increase the ionic channel density and the mechanical and thermal characteristics. Among the various filler materials, silica is the most widely used. Wang et al. [15] studied SiO_2_-Nafion composite membranes based on bifunctional SiO_2_ nanofibers. They analyzed the thermal stability, water uptake, dimensional stability, proton conductivity, and methanol permeability of bifunctional nanofibers with various polar groups. They found useful properties of thermal stability, water uptake, dimensional stability, and proton conductivity, because bifunctional SiO_2_ nanofibers form a fiber skeleton for dimensional stability and form continuous pathways for proton movement in the membrane. Kim et al. [16] synthesized core-shell-structured sulfonated SiO_2_-Nafion composite membranes. They proposed that strong hydrophilicity and additional sulfonic acid groups that were uniformly distributed on the membrane contribute to proton conductivity. Other studies on SiO_2_-Nafion composite membranes have demonstrated better performance than that of Nafion [17,18]. Silica, phosphtungstic acid (HPW), and Nafion composite membranes were studied by Lin et al. [19]. In their study, the ionomer expanded and silica formed ordered arrays, and HPW was subjected to an electrostatic-induced orientation. With this composition, the proton transport pathways were well constructed in the membrane, and the proton conductivity was enhanced.

Many studies on inorganic filler materials/Nafion composite membranes show good proton conductivity under insufficient hydration, as shown in Table 1. This result may be related to the creation of an ionic cluster network. However, as was the case in our previous studies, it is difficult to propose a clear proton conductivity enhancement mechanism for the composite membrane.

In a previous study, Nafion-sulfonated SiO_2_ (SSA) composite membranes were synthesized and characterized. These membranes showed superior performance capabilities, compared with those of pristine Nafion, such as better proton conductivity, power density, and mechanical and thermal properties. Figure 1 illustrates the proton conductivity of 1 wt% Nafion-sulfonated SiO_2_ (SSA) and Nafion samples from an earlier study [24]. Under all relative humidity conditions, the Nafion-SSA composite membrane exhibited twice the proton conductivity of Nafion. At ambient temperature and 100% RH, the proton conductivities of the Nafion and Nafion-SSA composite membranes were 111 mScm^−1^ and 230 mScm^−1^, respectively. However, in many studies, including the previous work by the authors, there have been insufficient investigations of the morphological variations of the ionic channel distribution, which is crucial when attempting to analyze proton conductivity.

One of the important roles of PEMs is to function as proton conductors, with performance capabilities that are directly related to the morphology of the membranes. PEMs have a heterogeneous morphology; one example is the combination of a hydrophobic backbone with hydrophilic sulfonic acid groups. The key to proton conductivity is the ionic channel network, which is created by the negatively charged sulfonic acid groups and water on the membrane. Therefore, it is difficult to fully understand the proton-conducting mechanisms of these membranes.

Electrostatic force microscopy (EFM) is a widely accepted tool for analyzing surface electrical characteristics. EFM can be used to examine the surface charge distributions and dielectric constants of locally charged materials. This method can also measure long- and short-range forces simultaneously, such as the Van der Waals force and electrostatic forces, using interaction-induced tip vibration changes. Based on these interactions, the topography and surface charge distribution can be mapped simultaneously. With EFM measurements, the detailed surface charge distribution can be observed by the force gradient between the tip and sample surface. However, it is difficult to separate the surface charge values from the individual sources of the electrical interaction, because the surface charges are measured using only tip vibration changes.

Therefore, it is necessary to develop an analytical model to better understand the complicated net force between the tip and sample surface for EFM measurements. Numerous analytical models are capable of interpreting EFM measurements [25,26,27]. Mélin et al. [28] introduced an analytical model for estimating the stored charge value on a surface based on a nanosized parallel-plate capacitor. They assumed that the force gradient is due to the dipole–dipole interaction between the tip and stored charge. Based on this assumption, they proposed an analytical model to numerically calculate the stored charge. Furthermore, they extended this model to consider the tip and sample surface with other capacitor shapes. Han et al. [29] proposed a relationship between the local charge and the phase value from EFM. They assumed that the phase value from EFM was the result of the net force, due to local charges between the tip and sample surface. They showed that the phase value is proportional to the amount of local charge using low-density polyethylene (LDPE) coupled with the injected charge. They also studied local charge movements. In this regard, Shen et al. [30] studied monolayer graphene oxide (GO) sheets. They investigated the degree of reduction of these sheets using a parallel capacitor model based on EFM measurements. Based on their model, they derived the capacitive force based on the difference between the dielectric constants of graphene and mica.

In this study, we compared the morphologies of pristine Nafion and SiO_2_ (SSA)–Nafion composite membranes. More specifically, we attempted to understand the proton conductivity enhancements of SiO_2_ (SSA)–Nafion composite membranes with variations in the ionic cluster distribution. To do this, the topography and phase maps from EFM were measured for both membranes. In addition, the phase lag values of both membranes were characterized using histograms. To achieve a clear numerical understanding of the ionic channel distributions of the membranes, a numerical approximation model, which was derived from the principles of EFM using a PEM, was applied, with the phase lag value extracted from the EFM measurements. Using this method, more accurate and detailed quantitative ionic channel distribution evolutions of SiO_2_ (SSA)–Nafion composite membranes are provided.

## 2. Experiment

In this study, a Park Systems XE-150 was used for the measurements and a platinum-coated tip was used. Its resonance frequency, spring constant, and tip radius were 75 KHz, 0.2 N/m, and less than 25 nm, respectively. The morphologies and EFM images of both samples were simultaneously scanned at a 1 μm × 1 μm spatial resolution under room temperature and humidity conditions.

The SiO_2_ (SSA)–Nafion composite membranes were prepared according to the literature [24]. First, the SSA with ethanol was mixed with a Nafion ionomer. Second, it was sonicated for 4 h and then stirred for 12 h. Third, the SiO_2_ (SSA)–Nafion solution was dried in a vacuum oven at a constant temperature of 60 °C for 4 h. In addition, it was dried at 80 °C for 4 h and at 100 °C for 4 h. Next, it was boiled in a 1 M H_2_SO_4_ solution for 4 h; finally, the membranes were rinsed using deionized (DI) water and dried at 60 °C for 12 h. To assess proton conductivity enhancements with the ionic channel distributions of the sulfonate Nafion-SSA composite membranes, the membranes were comparatively studied using EFM. Two samples, namely pristine Nafion and Nafion-SSA composite membranes, were prepared. A pristine Nafion sample was prepared by recasting the ionomer. The Nafion-SSA sample was fabricated using 1 wt% SiO_2_. The measurement and characterization processes were performed accurately and systematically in several steps. First, both membranes were soaked overnight in DI water. The membranes were removed from the vessel immediately before the measurements. Second, both membranes were scanned at a frequency of 1 Hz and a sample bias voltage of 2 V. Third, for a more detailed analysis, the Nafion-SSA composite membrane was scanned while sweeping the bias voltage from −2 to 2 V. Fourth, phase and morphology maps were created simultaneously. Finally, based on the phase map of the membrane surface, the phase shift variations over the membrane were characterized, using a numerical approximation model (NAM) proposed in an earlier work by the authors [31].

The development of NAMs is one way to understand the EFM signals from PEM measurements. The focus of this study is similar to that of Shen et al. [29] Their study aimed to quantitatively understand locally charged areas encompassed by non-conducting areas by using EFM and a well-agreed simple charged area. This model was not suitable for PEMs because it only applied to clearly separated non-conducting and conducting area materials. In PEMs, it is difficult to classify non-conducting and conducting domains on a microscopic scale. In PEMs, a few nanometer-sized ionic clusters, which consist of a pathway of protons, are randomly scattered non-conducting backbones. The tip of a PEM is larger than that of the ionic clusters. Thus, the EFM signal includes signals from the ionic cluster and the backbone. In addition, ionic clusters are created by hydration and are connected to each other, creating an ionic channel network and an ionic structure, which is the distribution of the ionic cluster network on the membrane changes. The protons then move through this network. Thus, the ionic structure of the membrane is changed by hydration.

To understand the ionic structure of a PEM, this condition must be considered. When a tip is placed on the PEM surface, the capacitive force between the tip and PEM simultaneously measures both the electrical forces from the induced electrical force from the backbone and the electrical force from the free charge, including protons. The analysis model based on EFM must include the influence of two electrical forces. In the NAM, the two electrical forces are based on two assumptions: First, the nano-sized capacitor is creased by a conductive tip and the PEM surface, and the nano-sized capacitor is assumed to be a parallel plate. Second, the electrostatic interaction between the conducting tip and the PEM surface is the net force of the induced electrostatic force, owing to the polarization of the backbone that is due to an external force and the columbic force that is due to free charges, such as protons and negatively charged sulfonic acid groups. Using the NAM, the surface charge distribution of the PEMs can be approximately calculated and the ionic structural change of the PEMs due to hydration can be numerically determined. When the series of EFM data is measured under a continuous hydration rate change, the change in the ionic structure evolution can be analyzed. In addition, the ionic structure of the membranes may be quantitatively analyzed, not only in the Nafion-SSA composite membrane, but also in other PEMs.

The use of the NAM is one way of determining the electrostatic interaction between the tip and the sample surface under EFM. In addition, the NAM can quantitatively characterize the electrical charge distribution on the sample surface, using the phase lag map from EFM. To measure the electrical charge distribution on the sample surface, a bias voltage was applied between the tip and sample surface. The surface charge distribution on the PEMs was measured using the same method. However, it is not easy to characterize the surface charge distribution compared with that for another conducting surface. To analyze the charge distribution, a capacitance force is typically assumed to exist between the tip and the sample surface, and is therefore modeled [22]. When using NAM, modeling is performed by assuming a nano-sized capacitor with a dielectric slab, which consists of a conducting tip, the membrane, and the conducting sample holder. If the insulating membrane surface is uniform, only a capacitive force is applied between the tip and the sample surface. Thus, an attractive force is always applied between the tip and the sample surface under positive or negative bias voltages. If electrical charges, which are called free charges, exist on the membrane surface, the electrostatic force is the sum of the capacitive and electrostatic forces, due to the free charges. Thus, the electrostatic interaction between the tip and the sample surface can be determined by the local magnitude of the force and changes in polarity.

## 3. Experimental Result

Figure 2a,b shows the topographies of the pristine Nafion and Nafion-SSA composite membranes, respectively. The color bar on the left side of the image is the height scale of the membrane according to the color depth. None of the membranes showed remarkable morphological characteristics. In the pristine Nafion case, the surface was curved to a few tens-of-nanometers in size, and the height varied within 1 nm, as shown in the line profile at the bottom of Figure 2a. The Nafion-SSA composite membrane exhibited a rough surface. The bulged area was a few hundred nanometers in size and was more than ten times larger than that of pristine Nafion; the height variation was a few tens-of-nanometers, as shown in the line profile of the Nafion-SSA composite membrane surface. This indicated that the Nafion-SSA composite membrane surface became rougher because of the added sulfonated silica. A more detailed characterization of the RMS roughness of both membranes was performed and itwas determined as follows.
(1)$$R_{q}=\sqrt{\frac{\sum^{\;}{\left(x_{i}-\bar{x}\right)}^{2}}{N}}$$
where *x*_i_, $\bar{x}$, and *N* are the height of a single pixel, the mean height, and the number of pixels, respectively.

With this calculation, the RMS roughness levels of the pristine Nafion and Nafion-SSA composite membranes were found to be 0.29 nm and 12.1 nm, respectively. These results implied that the filler materials that were added to the Nafion ionomer induced structural changes, such as an enlarged surface area. Owing to the relatively large and strong hydrophilic surface area, the water content on the membrane surface may increase and affect performance enhancement. Furthermore, the addition of sulfonic acid groups to the filler materials may also increase the number of ionic channels, as indicated in many earlier studies. This was also closely related to the increased performance capabilities of these membranes.

To analyze the surface charge distribution of both membranes, their phase maps were measured by applying −2 V and 2 V sample bias voltages. Figure 3 shows the results obtained when using −2 V. The color bar next to the images indicates the phase lag value on the surface. The phase lag value on the surface is related to the degree of surface charge. Thus, the phase map shows the surface charge distribution. Both membranes showed different features over the entire surface, compared with the simultaneously created topographies, as shown in Figure 2. For pristine Nafion, the phase lag value of the entire surface was positive and uniform. The positive phase lag value implied that a repulsive force was applied between the tip and sample surface. The phase value varied by less than ten degrees, as shown in Figure 3a,c, implying a uniform surface charge distribution for the pristine Nafion specimen. This result may have been affected by uniformly distributed surface charges. For the Nafion-SSA composite membrane, the phase value on the surface varied significantly from +150° to −100°, as shown in Figure 3b,d. The phase value of the overall surface area exceeded 50°, as indicated by the bright yellow area in Figure 3b. The repulsive force between the tip and sample surface was dominant in this area. In an area with a positive phase value, the value varied within a few tens of degrees. The black dots on the Nafion-SSA composite membrane stemmed from the negative phase value, which resulted from the attractive force between the tip and sample surface. This result implied that the surface charge distribution was not uniform. Therefore, more detailed calculations of the mean phase value of both membranes were undertaken, with corresponding outcomes of 40° and 90°, indicating that the surface charge value of the Nafion-SSA composite membrane was greater than that of the pristine Nafion membrane. However, it was very difficult to determine the increased ionic channel density on the surface because the net force on the surface is the sum of the ionic clusters, the charge induced by the applied bias voltage, and the charge due to silica.

Figure 4 shows the phase distribution of both membranes when using a bias voltage of 2 V. The overall membrane surface showed a positive phase value, indicating that the dominant interaction between the tip and membrane surface was a repulsive force. The phase value varied from 36° to 45°, as shown in Figure 4a. The variation in the phase value under a bias voltage of 2 V was similar to that obtained when a sample bias voltage of −2 V was applied. The Nafion-SSA composite membrane showed a more obvious phase separation, compared with the separation at a bias voltage of −2 V. The tiny negative-phase area (dark dots in Figure 4b) denotes the exposed image. Moreover, a large negative-phase area was observed at the middle-left position in the image. In the negative phase area, an attraction force was applied between the tip and the sample surface. Most of the area in the image had a positive phase value that varied between 50° and 100°, as shown in Figure 4d. This indicated that the interaction between the tip and the sample surface was relatively weak, compared with the interaction at a bias voltage of −2 V.

To quantitatively assess the interaction between the tip and the sample surface due to the surface charge distribution, a histogram was calculated from each image. Figure 5a,c presents histograms of pristine Nafion under −2 V and 2 V sample bias voltages, respectively, and both histograms showed a normal distribution. The peak values and FWHM of the pristine Nafion sample with the application of a −2 V bias voltage were 41° and 2.4°, respectively. For the pristine Nafion sample with a 2 V bias voltage, the peak value was 37° and the FWHM was 2.0°. The FWHM outcome was relatively similar to that of the pristine Nafion and indicated that the surface charge distribution was uniform during the overall interaction with the tip. A difference of a few degrees between the −2 V and 2 V bias voltage conditions indicated that a weak and uniform interaction change between the tip and the sample surface may have occurred because of the change in the sample bias voltage from −2 to 2 V. Figure 5b,d presents histograms of the outcomes when bias voltages of −2 and 2 V were applied to the Nafion-SSA composite membrane. In Figure 5b, normal distribution-like features appear, as in the case of pristine Nafion. However, negative phase values and extremely small peaks were observed near −100° and −150°, respectively, as shown in Figure 4b. In Figure 4d, clear positive and negative peaks appear with their shapes in each case, showing a normal distribution. A normal distribution, which has a positive value, is reflected by the bright-colored area on the membrane, as shown in Figure 3b and Figure 4b. A peak with a negative value was induced, as indicated by the dark colored dots and the area on the membrane, as shown in Figure 3b and Figure 4b. The peak value and FWHM of the Nafion-SSA composite membrane with the application of −2 V were 89° and 49°, respectively. The positive and negative peak values corresponded to 64° and −126°, respectively. The FWHM outcomes of the positive and negative peaks were 48° and 28°, respectively. For a bias voltage of 2 V, the peak value became 28% lower than that at a bias voltage of −2 V, although the FWHM did not change. A relatively large attractive force between the tip and the sample surface was observed when a bias voltage of 2 V was applied. This result implied that the charges on the membrane surface had locally different polarities and density characteristics.

Comparing the pristine Nafion (Figure 5a) and Nafion-SSA composite membranes (Figure 5b) with a sample bias voltage of −2 V, the peak and FWHM values were very different. The peak value of the Nafion-SSA membrane was two times larger than that of pristine Nafion, indicating that the electrostatic force between the tip and sample surface of SiO_2_-Nafion was much stronger than that of pristine Nafion. The FWHM of the Nafion-SSA was 20 times larger than that of the pristine Nafion samples and showed a non-uniform surface charge distribution. The weak negative phase value in the histogram of the Nafion-SSA composite membrane represented a small area, indicating a reciprocal interaction between the tip and the sample surface, as well as a non-uniform surface distribution. The histograms of the pristine Nafion (Figure 5c) and Nafion-SSA composite membranes (Figure 5d) under a bias voltage of 2 V showed features similar to those obtained when a bias voltage of −2 V was applied to the membranes. The peak value and FWHM differences between both membranes were larger by roughly two times and 20 times, respectively. Unlike the case when a sample bias voltage of −2 V was applied, a significant difference between the two membranes was that a negative peak clearly arose in the histogram, originating from the area with negative phase values in the image. This indicated that the surface charge distribution of the Nafion-SSA composite membrane under a 2 V bias voltage was non-uniform and that charges with different polarities also existed.

## 4. Analysis

To characterize the ionic channel distribution on the membrane from the EFM measurements, analyses of both membranes were conducted using the NAM, which was derived in a previous study. [31] This analysis can also provide an understanding of the electrochemical interactions between the tip and surface. Equation (1) determines the force gradient on the surface as follows:(2)$$\Delta f\sim -\frac{F^{\prime }}{2k}f_{0}=-\frac{\pi \varepsilon_{0}R_{tip}^{2}}{2k{\left(z+\frac{t}{\varepsilon_{r}}\right)}^{3}}f_{0}V^{2}+\frac{Q_{free}}{4kz^{3}}R_{tip}^{2}f_{0}V$$
where *k, R_tip_, f_0_, ε_r_, ε_0_, t, z, V,* and *Q_free_* are the spring constant of the cantilever, radius of the tip, resonant frequency, relative permittivity, permittivity in vacuum, membrane thickness, distance between the tip and sample surface, bias voltage, and the free charge on the membrane surface, respectively. The first term in Equation (2) indicates the force between the induced charge on the membrane surface due to the bias voltage and the tip. The second term in Equation (2) represents the force between the free charges on the surface and the tip. Thus, the frequency shift includes electrostatic forces due to the induced and free charges. Equation (2) describes the phase lag, which is derived from the multiplication of the amplitude of the cantilever and the frequency shift. Thus, the phase lag also includes induced and free charges. A positive phase lag value indicates that the second term in Equation (2) is dominant, while a negative phase lag value indicates that the first term in Equation (2) is dominant.
(3)$$\Delta \varphi \sim A\Delta f=-\frac{\pi \varepsilon_{0}AR_{tip}^{2}}{2k{\left(z+\frac{t}{\varepsilon_{r}}\right)}^{3}}f_{0}V^{2}+\frac{Q_{free}A}{4kz^{3}}R_{tip}^{2}f_{0}V$$

Figure 6 presents the electrostatic configuration of the tip and the pristine Nafion surface when applying negative (Figure 6a) and positive (Figure 6b) bias voltages. In the case of negative sample bias voltage, the conductive sample holder was negatively charged and the tip was positively charged. Positive charges were induced at the bottom of the Nafion membrane, while negative charges were induced on the other side. A water layer existed on the Nafion surface because it was measured under fully hydrated conditions in the ionic channels. As illustrated by the circle in Figure 6a, SO_3_H, SO_3_^−^, and H^+^ existed. While the Nafion surface was scanned under a negative sample bias voltage, hydrolysis occurred in the water layer and protons were generated. Due to the excessive and instantaneous generation of protons, the surface free charge was larger than the induced charge. Thus, the second term in the NAM was dominant and a repulsive force was applied between the tip and the sample surface. This interaction was due to the protons from the hydrolysis, and therefore it was difficult to characterize the ionic channel distribution on the surface.

Under a positive sample bias voltage, the charges on the tip and the sample holder were negative and positive, respectively. The top and bottom of the Nafion membrane were positively and negatively charged, respectively. In this case, hydrogen was produced by hydrolysis. In ionic channels, negatively charged sulfonic acid groups interacted with the negatively charged tip. The electrostatic force between the tip and the sample surface was a repulsive force, and the second term in the NAM was dominant. Thus, the phase value of Nafion under a positive bias voltage was directly related to the ionic channel network.

The Nafion-SSA composite membrane surface had a different electrostatic configuration, as shown in Figure 7. Under a negative sample bias voltage, protons were produced by the hydrolysis of the water layer. In addition, the water layer on the membrane surface was in a colloidal staten with protons and silica particles under strongly acidic conditions, and an electrostatic double layer was created around the silica. The water layer was notable because of its positive charge. The phase lag value on the membrane was mostly positive, indicating that a repulsive force was applied between the tip and sample surface. Thus, the second term in NAM was dominant, and the free charges in this case were protons and the positively charged water layer. Under the same conditions, the electrostatic force must be larger than that of Nafion. This was proven by comparing the phase lag values of both membranes, and this comparison indicated that the value for the SiO_2_-Nafion composite membrane was two times larger than that of pristine Nafion. However, the difference in the phase lag value between the two membranes was related to the ionic channels on the membrane, and did not provide meaningful information about the electrical structures of the membranes.

Figure 7b shows the electrostatic configuration of the Nafion-SSA composite membrane at a positive sample bias voltage. Hydrogen from electrolysis was exposed to the membrane surface. Thus, protons from the ionic channels diffused into the membrane surface and the ionic channels continued to contain SO_3_^−^. The electrostatic force between the tip and sample surface was the net force of the silicates and SO_3_^−^. Thus, the phase lag value under a positive bias voltage was directly related to the ionic channel distribution on the membrane surface. The repulsive force that appeared as a positive phase lag value indicated highly dense ionic channels and the attractive force that appeared as a negative phase lag value indicated low-density ionic channels or non-ionic channels.

The phase lag value of the membrane under a positive sample bias voltage was used to characterize the surface ionic configuration of the membrane. The ratio of the relative permittivity to the membrane thickness in the first term of the NAM originated from the membrane. The thicknesses of both membranes were approximately 50 μm. The relative permittivity levels of the Nafion and SiO_2_-Nafion composite membranes were approximately 4 and 3.9, respectively. From the blind reconstruction approximation, the tip radius was estimated to be 20 nm. Both membranes exhibited similar values, which were extremely small. Because the first term of the NAM was small, it could be ignored. Under identical conditions, the second term of the NAM was proportional to the free charge Q_free_. Each phase lag value was proportional to the sum of the free charge over an area of 1.2 × 10^3^ nm^2^, as calculated from the tip radius.

Table 2 summarizes the peak value and FWHM for an applied positive sample bias voltage. In the Nafion, the peak phase lag value and the FWHM were approximately 37° and 2°, respectively. The small FWHM value indicated that the ionic channels on the membrane were uniform. In the Nafion-SSA composite membrane, the positive and negative peaks of the phase lag values were 64° and −126°, respectively. The FWHM values of the positive and negative phase lag values for the Nafion-SSA composite membrane were 47° and 30°, respectively. The clear separation between the positive and negative peaks indicated the distinction between the rich and poor areas of the ionic channels. In the positive peak area, the peak for SO_3_^−^ was larger than that of the positively charged water layer, implying that the ionic channel density exceeded the circumference.

The FWHM ratio and peak value of the composite membrane/Nafion membrane were 23.5 and 1.7, respectively. They were used as a comparison tool for pristine Nafion and composite membranes. In the negative peak area, the positively charged water layer was dominant, and SO_3_^−^ was rare or absent. This result indicated that SiO_2_ and Nafion were not uniformly synthesized. If SiO_2_ and the ionomer were uniformly combined, ionic channels in the area were uniformly created in the membrane. If the silica agglomerated, the creation of an ionic channel was obstructed. From this approximation, 10% of the measured area showed a negative phase lag value, as shown in Figure 4b. Silica agglomerated in this area, disturbing the creation of the ionic channel network. Moreover, the relatively high FWHM indicated that the ionic channel density distribution over the membrane was not uniform. It was difficult to directly compare the uniformity of the ionic channels in both membranes. However, the FWHM from the histogram showed the dispersion of the phase lag under the measured area, indirectly indicating the uniformity of ionic channels on the membrane surface. Thus, the FWHM of the SiO_2_-Nafion composite membrane/pristine Nafion samples showed an indirect difference in uniformity between both membranes, with a ratio of 23.5. From the negative peak and relatively high FWHM, it was concluded that the Nafion-SSA composite membrane had a relatively non-uniform ionic channel distribution on the membrane compared to the Nafion membrane. The ratio of the mean phase lag value is proportional to the ratio of the second term of both membranes, which is related to the mean ionic channel density.

The ratio of both membranes was 1.7, indicating that the mean value of the ionic channel density of the Nafion-SSA composite membrane was more than 1.7 times larger than that of pristine Nafion. This was one of the reasons for the high proton conductivity of the Nafion-SSA composite membrane, and this result agreed with the ratio of the IEC, which was 1.41 according to an earlier work [21]. To summarize the evolution of the ionic channel distribution of the Nafion-SSA composite membrane, sulfonated silica helped to increase the mean local ionic channel density, such that the proton conductivity of the membrane was increased. Considering the FWHM of both membranes, the overall proton conductivity of the Nafion-SSA composite membrane was higher than that of pristine Nafion, but the proton conductivity of the Nafion-SSA composite membrane was unstable.

## 5. Conclusions

In this study, the surface charge distributions of Nafion and sulfonated SiO_2_-Nafion composite membranes were measured using EFM and characterized using the NAM. The phase lag values, as determined by EFM, are the sum of various electrostatic forces between the tip and the sample surface. To understand the ionic channel distribution on the surface, it is important to characterize the force that affects the phase lag value. The use of the NAM is one way to approximate the net electrostatic force of a sample surface based on the EFM configuration and the morphological structure of PEMs. In this study, several significant results were obtained. First, the local ionic channel distribution was estimated using the NAM. By measuring the phase lag of a selected area, the value for each pixel includes the local ionic channel density. The ionic channel distribution on the membrane was numerically approximated from an interpretation of the phase lag value using the NAM. Second, the uniformity of the ionic channel distribution was estimated using the NAM, and the results showed that the ionic channel distribution of pristine Nafion was much more uniform than that of the SiO_2_-Nafion sample. Third, the overall ionic channel density on the surface can be estimated using the NAM. The ionic channel density of SiO_2_-Nafion is roughly 1.7 times greater than that of pristine Nafion.

## Figures and Tables

**Figure 1 polymers-14-03718-f001:**
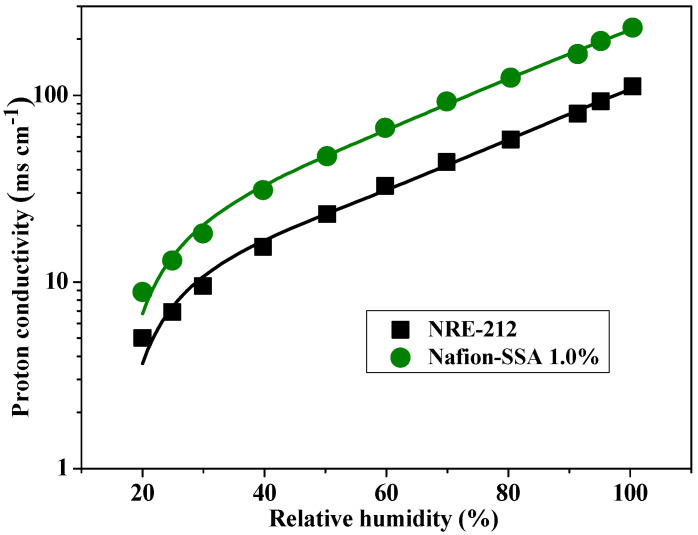
Proton conductivity of pristine Nafion and Nafion-SSA composite membranes [24].

**Figure 2 polymers-14-03718-f002:**
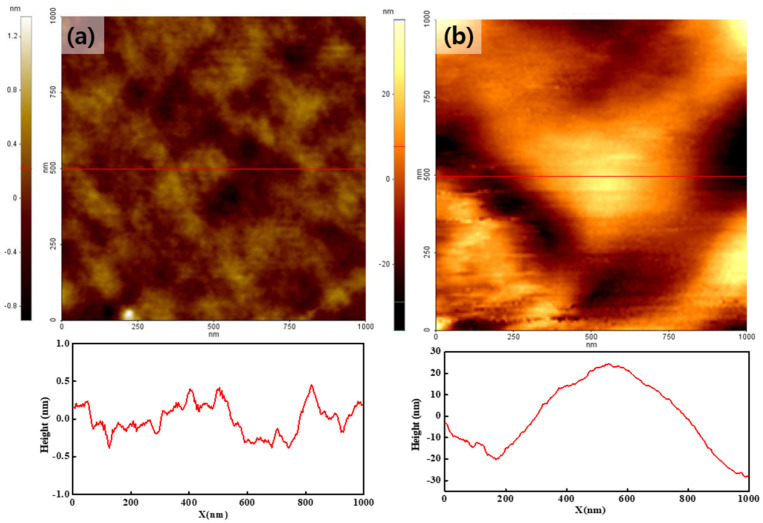
Topography of the pristine Nafion (**a**) and Nafion-SSA composite membranes (**b**).

**Figure 3 polymers-14-03718-f003:**
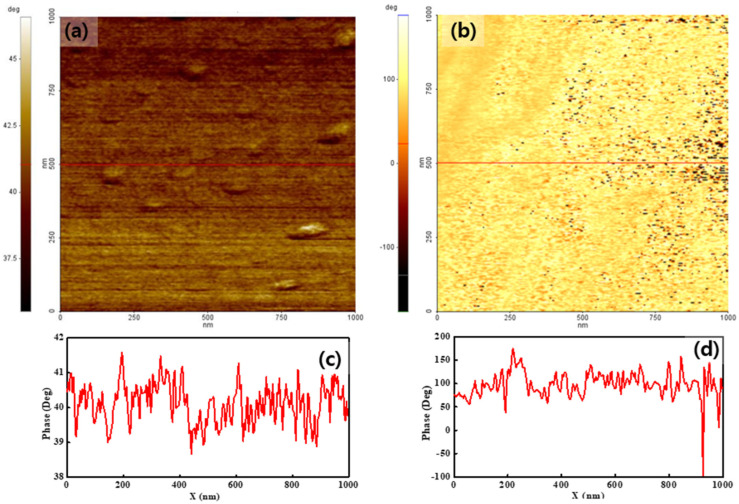
Phase images of pristine Nafion (**a**) with the line profile (**c**) and the Nafion-SSA composite membrane (**b**) with the line profile (**d**) under a −2 V bias voltage.

**Figure 4 polymers-14-03718-f004:**
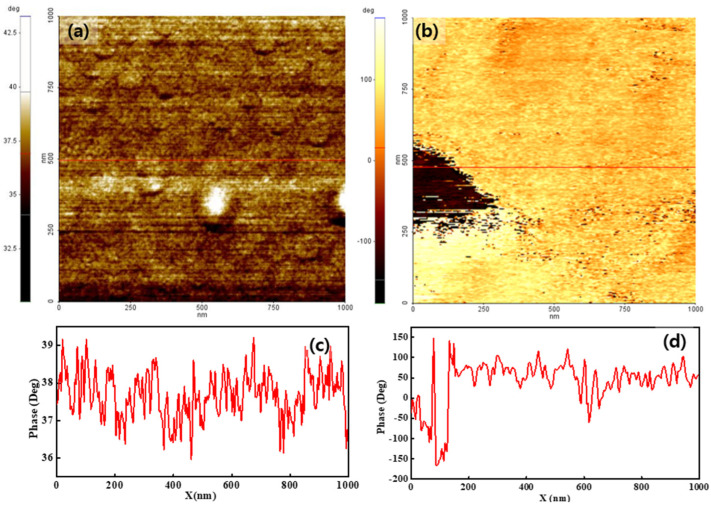
Phase images of pristine Nafion (**a**) with the line profile (**c**) and Nafion-SSA composite membrane (**b**) with the line profile (**d**) under a 2 V bias voltage.

**Figure 5 polymers-14-03718-f005:**
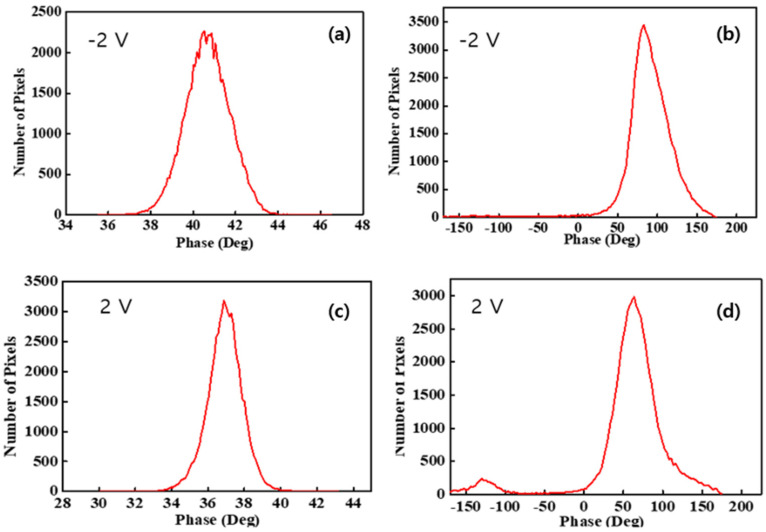
Histograms of membranes: (**a**) pristine Nafion under a −2 V sample bias voltage, (**b**) Nafion-SSA composite membrane under a −2 V sample bias voltage, (**c**) pristine Nafion under a 2 V sample bias voltage, and (**d**) Nafion-SSA composite membrane under a 2 V sample bias voltage.

**Figure 6 polymers-14-03718-f006:**
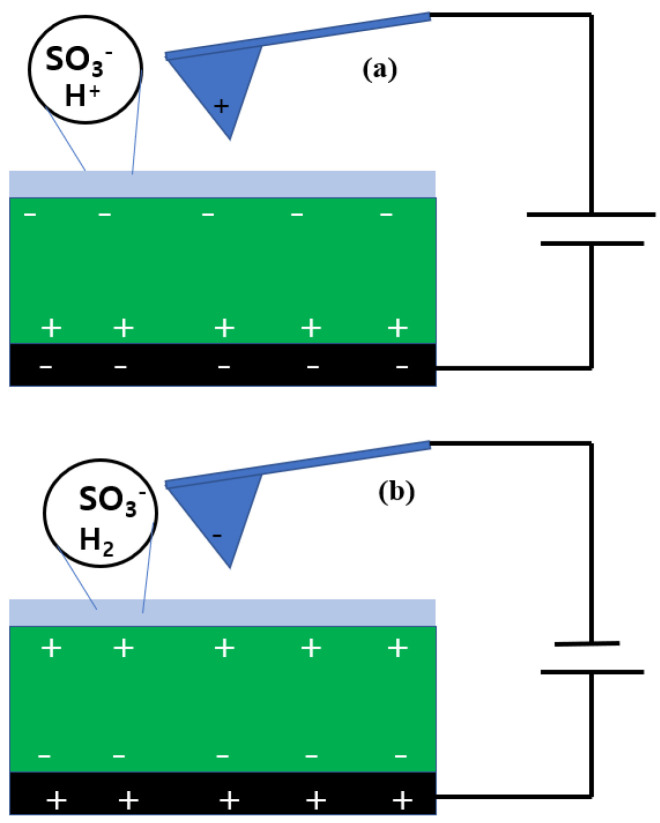
Electrostatic configuration of the tip and the membrane surface when negative (**a**) and positive (**b**) bias voltages were applied.

**Figure 7 polymers-14-03718-f007:**
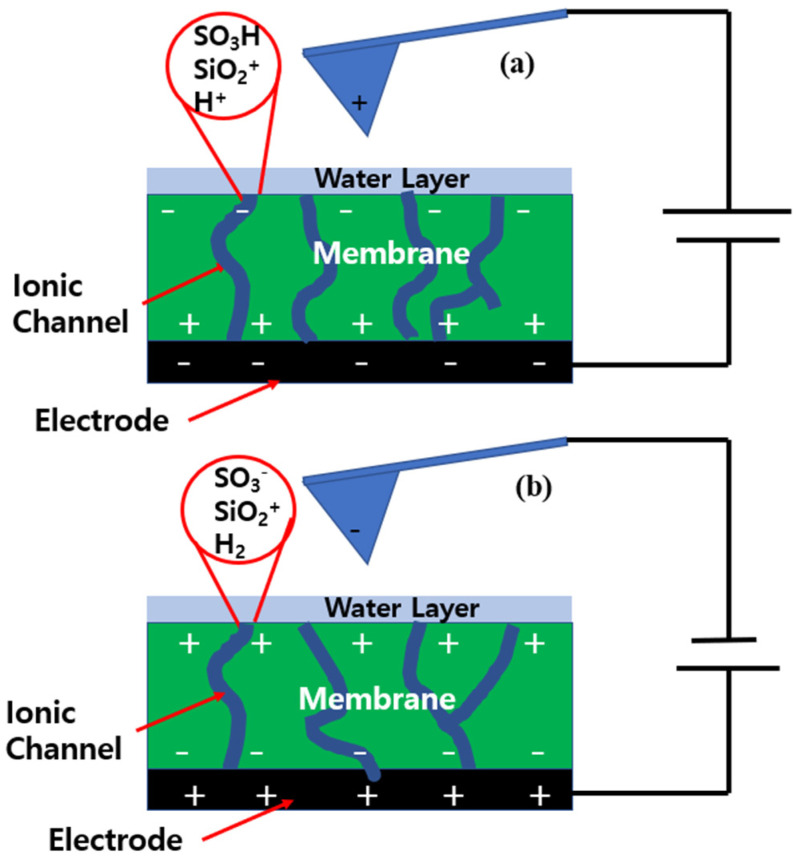
Electrostatic configuration of the tip and the Nafion-SSA surface when applying negative (**a**) and positive (**b**) sample bias voltages.

**Table 1 polymers-14-03718-t001:** Proton conductivity of proton exchange membranes.

	Conductivty (mS/cm)	Max Power Density (mW/cm)	Condition	References
Recast Nafion	90	900	100% RH, 80 °C	This work
	4.1	250	25% RH, 80 °C	This work
Nafion–SSA 1.0 wt%	8.8	454	25% RH, 80 °C	This Work
	200			This work
Silica dioxide nano paricle 5%	40		40% RH, 80 °C	[20]
Nafion/SGF-3	3	200	25% RH, 80 °C	[21]
Recast Nafion/SiO_2_/PWA	70		80% RH, 100 °C	[22]
SiO_2_ supported sulfated zirconia/Nafion	70	1 w	100% RH, 60 °C	[23]
Silfonated graphene oxide Nafion	130	900 mW	100% RH, 60 °C	[24]

**Table 2 polymers-14-03718-t002:** Phase lag value and FWHM of both membranes under the application of a positive sample bias voltage.

	Positive Peak	Negative Peak	FWHM of Positive Peak	FWHM of Negative Peak
Pristine Nafion	37°		2°	
Nafion-SSA composite membrane	64°	−126°	48°	26°

## Data Availability

Data are contained within the article.

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
