# Peer review of "Analysis of Ionic Domain Evolution on a Nafion-Sulfonated Silica Composite Membrane Using a Numerical Approximation Model Based on Electrostatic Force Microscopy"

_polymers, 2022, doi:10.3390/polym14183718_

Round 1

Reviewer 1 Report

The work «Analysis of Ionic Domain Evolution on a Nafion-Sulfonated Silica Composite Membrane by using a Numerical Approximation Model based on Electrostatic Force Microscopy» is devoted very interesting topic about proton conductivity enhancements of composite membranes for gas separators, electrode insulation, and proton conductors. In the introduction, the authors described in detail the necessity and importance of the study, presented the results of previous works, including their own, and formulated the purpose of this article. In the experimental part, the conditions for obtaining of the SiO2 (SSA)-Nafion composite membranes and the methods used are described. The main part of the article is successfully divided into experimental results and analysis. The figures are useful enough to understand the data obtained. All the main results obtained are summarized in the conclusion. The article can be improved by considering the following comments:

1. In the "experiment" section, there is no description of the equipment for measurements.

2. Figure 6 has So3-. What's this?

3. Table 3 should be simplified and given in plain text.

Author Response

First of all I appriciate your accurate and detailed review for my manuscript.

My answer about your comments are listed below.

  1. I added sentence of equipment for measurements in the manuscipt.
  2. I fixed Figure 3.
  3. The table 3 wrote in plain text.

Reviewer 2 Report

The manuscript reported the quantitative analysis of the proton conductivity enhancement for Nafion-SSA composite membranes with variations in the ionic channel distribution compared to pristine Nafion. It is important to characterize the proton transport mechanisms of membranes, especially the morphological structure. EFM is the best tool for characterizing the ionic structure, but it is difficult to analyze the data from the measurements. In this study, the data from the EFM measurements are analyzed by the NAM. The findings are meaningful and imply that the NAM is a proper tool for quantitative assessments of PEMs.

I consider the content of this manuscript will definitely meet the reading interests of the readers of the Polymers journal. However, there are certain English spelling and grammar issues, and also the discussion and explanation should be further improved.

Therefore, I suggest giving a minor revision and the authors need to clarify some issues or supply some more experimental data to enrich the content. This could be comprehensive and meaningful work after revision.

Detailed comments can be found in the PDF file.

Author Response

First of all I appreciate your accurate and detailed review of my manuscipt.

My answer about your comments are listed below.

  1. I reduced the manuscript within 200 words.
  2. I did english editing from the professional editing company.
  3. I added recommended key words on the manuscript.
  4. I wrote brief sentence for explain chemical structure of Nafion on the manuscript.
  5. I corrected Nafion 212 to recast Nafion.
  6. I deleted the sentence.
  7. I fixed Figure 3.
  8. I added basic introduction paragraph of NAM on the manuscipt.
  9. I changed starting of the introduction.